# Latent Tuberculosis Treatment among Hard-to-Reach Ethiopian Immigrants: Nurse-Managed Directly Observed versus Self-Administered Isoniazid Therapy

**DOI:** 10.3390/tropicalmed8020123

**Published:** 2023-02-16

**Authors:** Hashem Bishara, Manfred Green, Amer Saffouri, Daniel Weiler-Ravell

**Affiliations:** 1Tuberculosis Clinic and Pulmonary Division, Galilee Medical Center, Nahariya, and Azrieli Faculty of Medicine, Bar-Ilan University, Safed 5290002, Israel; 2School of Public Health, Faculty of Social Welfare and Health Science, University of Haifa, Haifa 3498838, Israel; 3Tuberculosis Clinic and Internal Medicine, Nazareth Hospital, and Azrieli Faculty of Medicine, Bar-Ilan University, Safed 5290002, Israel; 4Former Director of Pulmonary Division and Tuberculosis Clinic, Carmel Medical Center, Haifa 3436212, Israel

**Keywords:** latent tuberculosis, immigrants, treatment, preventive therapy, reception centers, Ethiopian, cost, treatment completion

## Abstract

Background: The treatment of latent tuberculosis infection (LTBI) among high-risk populations is an essential component of Tuberculosis (TB) elimination. However, non-compliance with LTBI treatment remains a major obstacle hindering TB elimination efforts. We have previously reported high treatment compliance with nurse-managed, twice-weekly, directly observed Isoniazid treatment (DOT) for LTBI among hard-to-reach Ethiopian immigrants (EI’s). Objectives: to compare rate of completion of treatment, cost, and major adverse drug events with daily self-administered Isoniazid treatment (SAT) to nurse-managed Isoniazid DOT among hard-to-reach EIs. Materials and Methods: We conducted a retrospective study and compared self-administered LTBI treatment outcomes among EIs housed in reception centers during 2008–2012 to EIs treated with DOT. Results: Overall, 455 EIs were included (231 DOT, 224 SAT) in the study. We found no significant difference in treatment completion rates between the two groups (93.0% DOT vs. 87.9% SAT, *p* = 0.08). However, cases of grade III, drug-induced hepatitis were significantly fewer and treatment costs were significantly lower with the nurse–managed DOT compared with SAT (0% vs. 2.2%, *p* = 0.028, 363 vs. 521 United States Dollars, *p* < 0.001, respectively). Conclusions: Nurse-managed, twice-weekly DOT among hard-to-reach EIs housed in reception centers had less severe drug-related adverse events and reduced treatment cost compared with daily isoniazid SAT, yet we found no significant difference in treatment completion between the two strategies in this population.

## 1. Introduction

Tuberculosis (TB) is a major cause of morbidity and a leading cause of death worldwide [1]. According to the 2022 WHO report, the global estimated number of TB deaths in 2021 was 1.4 million among HIV-negative people and 187 thousand among HIV-positive people [1]. The WHO “end the global TB epidemic” initiative aims to achieve a 50% reduction in the TB incidence rate by 2025 [2]. This goal might be achievable through enhanced screening for active and latent TB (LTBI) among high TB-risk groups and by providing efficient and timely treatment [1,2,3]. 

Immigrants from high TB-burden countries to low-incidence TB countries have a higher LTBI and TB burden compared with the native population in the host country [4,5,6,7]. Therefore, treating LTBI in such high-risk populations is a key component of the WHO end TB strategy.

While isoniazid was the mainstay for LTBI treatment since the 1960s, the Rifamycin-based regimens became the preferred option according to the 2020 guidelines [8,9]. This was mainly due to a higher treatment-completion rate compared with isoniazid monotherapy. However, management of LTBI has more to it than achieving a high treatment-completion rate. It is a multistep process that begins with identifying high TB-risk populations for LTBI screening, detecting individuals with LTBI, initiating treatment for LTBI among infected subjects, and finally ensuring treatment completion [10]. Although treatment completion is an important step in the LTBI cascade of care, more attention has been focused on treatment completion once LTBI is initiated than on earlier steps in the cascade. A recently published study evaluating the LTBI care cascade among public health clinics in the United States found that of all participants tested positive for LTBI, only 42% began therapy, of whom 76% completed treatment [10]. A systematic review and meta-analysis concerning initiation and completion of treatment for LTBI in migrants globally reported that 69% of immigrants testing positive for LTBI began treatment, of whom 75% completed treatment, resulting in a pooled estimate for LTBI treatment completion of 52% among eligible candidates [11]. Therefore, initiation and completion of treatment for LTBI is far from being adequate, regardless of the treatment regimen used.

There are currently four treatment options for LTBI: daily isoniazid for 6–9 months, a three-month regimen of daily isoniazid plus rifampicin, four months of daily rifampicin, or a three-month regimen of daily isoniazid plus rifampicin [8,9].

Although isoniazid is not the preferred choice for LTBI treatment, there are certain advantages with isoniazid treatment compared to rifamycin-based LTBI treatment options. Isoniazid’s low cost makes it the preferred option in low-income countries. The cost of isoniazid medication required to complete a six-month course of daily treatment is ~13 USD. Isoniazid can be prescribed twice weekly, making directly observed treatment possible, has fewer drug-drug interactions, and does not affect contraceptive medication pharmacokinetics. It also has a low incidence of drug adverse effects among younger patients [12,13,14]. Furthermore, there are some concerns that excessive use of the rifamycin-containing regimens could lead to acquired drug resistance, particularly if active TB diagnosis is missed [15,16]. The implications of rifamycin resistance on TB treatment cannot be overemphasized.

The main drawback of isoniazid treatment is its lower treatment completion rate compared to the other regimens. Non-adherence to LTBI treatment, particularly among high-risk populations, carries the risk of converting to active TB and perpetuation of the infectious cycle [17]. 

We have used the nurse-managed, twice-weekly isoniazid treatment (DOT) as a standard regimen for LTBI treatment among Ethiopian immigrants (EIs) living in reception centers in the city of Zefat for the last three decades. This was possible due to cooperation with the Ministry of Health nursing staff in Zefat. We have observed high rates of LTBI treatment completion with this strategy among immigrants from Ethiopia to Israel [18]. However, due to administrative issues, we could not use the same strategy and provide DOT in reception centers in the Yizrael subdistrict where we had to revert to a self-administered treatment (SAT) approach.

The primary objective of this study was to examine LTBI treatment completion rates with SAT, compared to DOT among hard-to-reach EIs housed in reception centers. The secondary objectives were to compare the prevalence and severity of drug-related adverse events and treatment cost between the two treatment strategies. 

We hypothesized that DOT treatment would be superior to SAT in terms of treatment completion, severity of drug related adverse effects, and cost among hard-to-reach EIs living in reception centers.

## 2. Materials and Methods

### 2.1. Settings

TB incidence among the native population of Israel has been steadily declining during the last two decades, to a 1/100,000 person-year in 2010 [19]. The majority (85%) of TB patients in Israel are foreign born and were infected with TB before their arrival in Israel [20]. The number of TB cases in Israel increased in the early 1990s, mainly due to an influx of immigrants of Jewish descent from high-burden countries, Ethiopia in particular. Upon arrival in Israel, EIs are housed in reception centers and granted citizenship, which includes full health care benefits. They undergo screening for active and LTBI before their arrival in Israel. The screening consists of a chest X-ray and tuberculin skin test (TST). The TST was conducted using the Mantoux method, by intradermal injection of 0.1 mL 5U purified protein derivative to the forearm volar surface. Immigrants with a TST measuring ≥10 mm or ≥5 mm for close contacts of active TB cases or with an abnormal chest X-ray were referred shortly following their arrival in Israel to TB clinics for further evaluation, which may have included sputum cultures, chest CT, and bronchoscopy as indicated. All TB-related medical management in Israel is provided by a TB specialist at specific TB clinics, nine in number, serving the entire population of Israel. 

Reception centers in Zefat, northern Israel, began operating in the early 1990s, and the nursing staff of Zefat subdistrict health office provided medical services to the EIs residing in this center, performed TST, and delivered LTBI medication. With increasing numbers of EIs arriving in Israel, additional reception centers were set up as needed to accommodate the new arrivals. All centers were managed by a house administrator, all were funded and operated in the same manner, and all had a team of workers dedicated to the well-being of the residents in these centers. Such a center was established in the Yizrael subdistrict, but nursing staff were not recruited to deliver medical services to immigrants living at this center.

The Nazareth TB clinic is one of the nine regional TB clinics operating in Israel under the New National TB Control Program [18]. It serves the population of the northern district of Israel, which includes the Zefat and Yizrael subdistricts. The Nazareth TB clinic is located 40 km from the reception centers with no direct public transportation route available. EIs residing in the Zefat reception center were routinely treated with nurse-managed, directly observed, twice-weekly isoniazid treatment (DOT) for LTBI. However, EIs in the Yizrael reception center were treated with daily, self-administered isoniazid therapy (SAT) due to the absence of nursing staff as mentioned previously. Both groups were initially evaluated and further followed up at the Nazareth TB clinic by the same nursing staff and pulmonologist (HB) who made all the decisions regarding treatment initiation, discontinuation, and completion.

### 2.2. Design

This was a retrospective study comparing LTBI treatment outcomes of SAT with daily isoniazid to twice-weekly, nurse-managed DOT from 2008 to 2012 among EIs housed in reception centers.

The SAT regimen was a self-administered dose of 5 mg/kg isoniazid once daily for adults or 10 mg/kg once daily for children, with a maximal daily dose of 300 mg. 

The DOT regimen was as previously described [18]. A dose of 15 mg/kg isoniazid for adults or 20 mg/kg for children (900 mg maximal dose) was administered twice weekly. A public health nurse oversaw the ingestion of one weekly dose and questioned the patient regarding side effects, while the second weekly dose was self-administered. Treatment completion in the DOT group was defined as completion of 26 supervised isoniazid doses during nine months; in the SAT group, treatment completion was defined as the intake of at least 142 (80% of the 6 months’ designated doses) SAT doses during the nine months. Both groups were followed up by the same medical staff at the Nazareth TB clinic. 

Monthly physician follow-up visits were scheduled at Nazareth TB clinic for the SAT group to provide medication and check for side effects, whereas the DOT group had two physician follow-up appointments during the entire treatment period. Additional follow-up appointments were scheduled if needed. Cultural case-managers to address the special needs of these immigrants from Ethiopia were assigned to both groups, and direct bus transportation to the TB clinic was provided free of charge.

### 2.3. Inclusion Criteria

EIs housed in reception centers in Zefat and Yizrael testing positive for LTBI who were referred to the Nazareth TB clinic were included.

### 2.4. Exclusion Criteria

Individuals who were relocated during the treatment period, pregnant women, and those with HIV infection or baseline liver transaminase levels >3 times the upper limit of normal were excluded from the study.

### 2.5. Adverse Drug Events

Fatigue, malaise, headache, dizziness, and insomnia were defined as nonspecific adverse events. Abdominal pain or discomfort, nausea, and vomiting were defined as Gastro-Intestinal (GI) symptoms. Hepatitis was defined as the presence of abnormal (> 35 upper level of normal) liver transaminase levels. Hepatitis adverse drug events were graded in accordance with the National Cancer Institute Common Terminology Criteria for Adverse Events [21].

### 2.6. Cost Calculation

The total cost of treatment for each group was calculated as the sum of the costs of interventions in that group, according to Israel’s Ministry of Health tariffs. Treatment costs included the physician appointment fee, cost of isoniazid, nursing staff, transportation to the TB clinic, and blood tests (liver transaminase levels). Nursing cost was calculated as the cost of a part-time nurse (12 h weekly), during the entire treatment period.

The average treatment cost per person who completed treatment was calculated by dividing the total sum of treatment costs for each group by the number of persons who completed treatment. The costs are presented in United States Dollars (USD, conversion rate: 1 USD = 3.5 Israeli new shekels).

### 2.7. Statistical Analysis

Differences between the 2 groups were assessed via the χ2 test or Fisher exact test in the case of rare events for categorical variables (such as gender, side effects, treatment completion) and by the Mann–Whitney U-test for Continuous variables. Statistical analyses were conducted using SPSS, version 19 (Statistical Package for the Social Sciences, Armonk, NY, USA). *p* values < 0.05 were considered statistically significant.

## 3. Results

We examined 497 files of which 42 were excluded. The main reason for exclusion was relocation to another district; four women were excluded because of pregnancy. Overall, 455 EIs were included in the study, with 224 in the SAT group. The two group’s characteristics were similar in terms of age and gender (Table 1). 

The treatment completion rate was high in both groups (Table 1), though slightly lower in the SAT group than in the DOT group (87.9% vs. 93.0%, respectively, *p* = 0.08). The prevalence of drug-related adverse effects was high, with a total of 182 adverse events reported. Adverse drug events were significantly higher in the SAT group as compared to the DOT group. Drug-related adverse events were mainly Grade I. Overall, 14.2% of the participants had abnormal aminotransferase blood levels, mostly grade I (78.4%, liver enzyme levels 1–3 fold of the upper limit of normal). Five participants had grade III hepatotoxicity (liver enzyme levels 5-20-fold above the upper limit of normal), all in the SAT group. No clinical jaundice was observed, and no hospitalization occurred due to drug-related adverse effects during the treatment period. There were 29 permanent treatment terminations due to adverse drug events, which accounts for two thirds of treatment non-completion; 18 were in the SAT group (*p* = 0.23).

The cost of treatment per participant who completed treatment was significantly higher in the SAT group compared to the DOT (521 US$ vs. 363 US$, *p* < 0.001).

Treatment completion rate was higher among males as compared to females, but this was not statistically significantly different (Table 2). There was no significant difference in completion rate between the age groups. However, subjects who did not complete the treatment had 12.9 times the odds of adverse drug events as compared to those who completed the treatment.

## 4. Discussion

We found high treatment completion rates in both groups with no statistically significant difference between the two treatment strategies. However, there was a significantly higher prevalence of GI and nonspecific drug-related adverse events and grade III hepatitis in the SAT group compared to the DOT.

We did not find any significant difference in treatment completion related to gender or age group. However, participants who reported adverse drug events had 12.89 odds ratio to treatment non-completion compared to those who did not report adverse drug events. Finally, treatment cost was significantly higher for the SAT group compared to the DOT. 

LTBI treatment completion rates are usually reportedly lower with isoniazid treatment as compared with the shorter regimens. Horsburgh et al., reported a 47% 9-month, isoniazid regimen, treatment-completion rate at public and private clinics in the United States and Canada in 2002 [22]. Residence in a congregate setting and those starting the 9-month isoniazid regimen were the main risk factors for failure to complete treatment. 

Although we found no significant difference in treatment completion rates between DOT and SAT, other studies reported higher LTBI treatment completion with DOT compared to SAT. Heal et al. [23] examined LTBI treatment completion among 608 aboriginal people in British Columbia and found lower completion rates with SAT than with DOT (60.9% vs. 75.2%, respectively, *p* = 0.001). However, the patients made the choice of the treatment regimen they would receive, and the DOT group had a significantly younger average age compared with the SAT group. White et al. [24] compared treatment completion before and after a DOT approach was adopted and found significantly higher rates with DOT than with SAT (70.3% vs. 54.9%, respectively, *p* = 0.001). However, the two groups differed significantly in terms of ethnicity and length of stay in the United States. Our treatment completion rates were comparable to those reported by Schein et al. (91%) among 719 individuals, mainly foreign born, all notified with LTBI treatment by the Norwegian Surveillance System for Infectious Diseases in 2016 [25]. The authors concluded that DOT strategy and involvement of a TB coordinator had a significant effect on treatment completion among the foreign born.

It is worth noting that EIs in Israel have a different milieu compared to the Canadian and American groups [22,23,24]. They are granted citizenship upon arrival, housing and a monthly allowance, and are cared for by the reception center staff for the first two years following their arrival. This probably enhanced their adherence to treatment, regardless of the mode of treatment administration. It is possible that the EIs readiness to cooperate with the reception center staff and comply with the immigration regulations had a greater impact on their adherence to treatment than the treatment approach itself (DOT vs. SAT). 

There were 182 reported adverse events in both groups; 29 of these events were severe enough to warrant permanent treatment termination. These subjects were offered a 4-month rifampin treatment course, and most of them completed treatment. We have no clear explanation for the high prevalence of mild GI and nonspecific complaints in the SAT group. There were five EIs with grade III isoniazid hepatotoxicity in the SAT compared with none in the DOT group. The fact that no grade III hepatitis was identified in the DOT group may have resulted from early identification of drug-related hepatitis symptoms by the nurse supervising the DOT, which led to prompt treatment discontinuation. 

The cost of treatment in the SAT group was higher than in the DOT group mainly because of the added cost of more frequent physician follow-up appointments and, consequently, higher transportation cost, compared with the DOT group. The provincial nature of this precinct made transportation a daunting and costly task.

This study had a number of strengths. We examined identical populations in terms of ethnic and cultural background, immigration status, epidemiologic characteristics, housing milieu, and linguistic barrier. Both groups were treated by the same TB clinic staff. Both groups had Israeli citizenship status and received the same immigration benefits granted by the state of Israel. The immigrants’ allocation to different reception centers was random and related mainly to the availability of vacant apartments in these centers. 

This study has some important limitations. The retrospective study design and the relatively small number of subjects in each group compromised the study’s statistical power and may have masked otherwise significant differences in outcome. The settings of the study population make our results restricted to similar constellation and milieu. The fact that the number of scheduled physician follow-up appointments differed between the two groups, which had a major impact on treatment cost, is also an important limitation. Monthly physician follow-up appointments to members of the SAT group to confirm their adherence to treatment, check for side effects, and provide monthly medication, added to the cost of the treatment. These tasks were performed weekly by the nurse in the reception center in the DOT group, and thus, only two follow-up visits by a physician for the DOT group were needed. The 2.7% grade III isoniazid-related hepatitis in the SAT group underlines the need for frequent follow-up examination to timely detect adverse drug reactions and prevent progression to serious liver injury.

## 5. Conclusions

Nurse-managed isoniazid DOT for LTBI is an effective, safe, and cost-saving alternative to SAT among high-risk, hard-to-reach, recent immigrants housed in reception centers. Well-organized health services can achieve high LTBI treatment completion rates, and also minimize the attrition throughout the LTBI cascade of care regardless of the treatment regimen.

## Figures and Tables

**Table 1 tropicalmed-08-00123-t001:** Demographic characteristics and outcome measures of latent TB treatments among Ethiopian Immigrants in Israel.

	DOT N (%) N = 231 (100%)	SAT N (%) N = 224 (100%)	OR (95% CI)	*p*
Gender- Male	122 (52.8)	121 (54.0)	1.05 (073–1.52)	0.80
Age, Mean ± SD	29.4 ± 15.9	29.7 ± 16.6	---	0.83
Age group				0.45
<18	63 (27.2)	67 (29.9)	1.00 (reference)	
≥18–35	84 (36.4)	69 (30.8)	1.29 (0.81–2.07)	
>35	84 (36.4)	88 (39.3)	1.02 (0.64–1.60)	
Adverse drug events ***n*** (x% of number of patients in the group)				
Gastrointestinal	12 (5.2)	48 (21.4)	4.98 (2.56–9.66)	<0.001
Rash/Pruritus	5 (2.2)	6 (2.7)	1.24 (0.37–4.14)	0.74
Non-specific	16 (6.9)	30 (13.3)	2.08 (1.10–3.93)	0.02
Prevalence and severity of drug related Hepatitis ^‡^				
Grade I Grade II Grade III	10 (4.3) 4 (1.7) 0 (0)	41 (17.6) 5 (2.1) 5 (2.1)	4.95 (2.41–10.16) 1.30 (0.34–4.89) 11.60 (0.64–211.04)	<0.001 0.70 0.028
Treatment completion	215 (93.0)	197 (87.9)	0.54 (0.28–1.04)	0.08
Average cost USD	363	521	----	<0.01

SAT = self-administered isoniazid treatment; DOT = directly observed isoniazid treatment; ^‡^ = National Cancer Institute Common Terminology Criteria for Adverse Events v5.0, USD = United States Dollar.

**Table 2 tropicalmed-08-00123-t002:** Treatment completion by gender, age groups, and adverse drug events among Ethiopian Immigrants.

Treatment	Completed	Not Completed	*p*	OR (95% CI)
**Gender** Male Female	226 (93.0) 186 (87.7)	17 (7.0) 26 (12.3)	0.06	0.54 (0.28–1.02) 1.00 (reference)
**Age group** **18>** **≥18–35** **>35**	121 (93.1%) 137 (89.5%) 154 (89.5%)	9 (6.9%) 16 (10.5%) 18 (10.5%)	0.51	1.00 (reference) 1.57 (0.67–3.68) 1.57 (0.68–3.62)
**Side effects** Yes No	84 (20.4) 328 (79.6)	33 (76.4) 10 (23.6)	<0.001	12.89 (6.11–27.20) 1.00 (reference)

## Data Availability

The data that support the findings of this study are available from the corresponding author (hashenb@gmc.gov.il) with the permission of the Nazareth Hospital.

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
