# Peer review of "Latent Tuberculosis Treatment among Hard-to-Reach Ethiopian Immigrants: Nurse-Managed Directly Observed versus Self-Administered Isoniazid Therapy"

_tropicalmed, 2023, doi:10.3390/tropicalmed8020123_

Round 1

Reviewer 1 Report

Dear colleagues,

The following items should be corrected:

-          The aim of the study is not fully understandable and consistent with the title of the article – as it should be defined straightforwardly, otherwise upon review of the whole article it is still not clear what conclusions might be expected.

-          -              The structure of the abstract should be corrected and should include: background; objective; materials and methods; results; limitations, and conclusions

-          The keywords should be more than five.

-    TB epidemiology data based on the WHO Reports 2020- 2021 should be included in the introduction. 

-  References are not up to date.

-          The aim of the study needs-assessment to make it consistent with design of the study.

-          Inclusion criteria not clear. What immunologic methods were used by colleagues the study (Mantoux test 2TU, ELISPOT and QTF-plus/QFR-Gold)? It should be defended in the methods of the study.

- Characteristics of patients should be presented chapter Materials and Methods.

-          Was X-ray or CT examination used in persons with positive results of immunologic tests?

The results of the study is not clear.

-      Conclusion is not clear. In Conclusions chapter, general concepts and information are given and it is not clear practical relevance of this study. Please formulate the conclusions more clearly and precisely with practical relevance clarification. 

-              All these concerns should be well addressed to consider this manuscript suitable for publication.

Author Response

We wish to thank the reviewers for their valuable comments.

Reviewer 2 Report

General comments 

It is a pleasure to review the manuscript titled ‘Latent Tuberculosis treatment among hard-to-reach Ethiopian immigrants: Nurse-managed directly observed versus self-administered isoniazid therapy’. This topic is relevant because it looks at a major public health problem affecting a vulnerable population in Israel. However, the quality of the manuscript can be improved further after addressing the following comments

Specific comments

Abstract

Please add information on the study population and duration of the study. 

Introduction 

The introduction is long with disjointed ideas. For example, in lines 35 to 39, the authors described the burden of latent TB infection. Following this, the authors provided details to give details of LTBI treatment and come back with other sentences describing latent TB in Israel in lines 59 and 60. The authors should improve on the chronological organization of the sentences. 

Please add a reference after the sentence ‘However; there are some 48 advantages of using isoniazid compared with other LTBI treatment options, mainly its low 49 cost, the option for a twice weekly dosing, low incidence of drug adverse effects at younger 50 age, and avoiding the risk of drug resistance following rifamycin-containing LTBI treatment regimens’ and remove the semi-Colum after ‘However’

The sentences on lines 69 to 86 are not required in the introductory section of this paper. They can be included in the methods section

Generally, an introductory section should determine the burden of evidence available on the subject matter, identify the gaps in knowledge and proffer an aim to address the gap. The authors did not use this approach at all. 

Materials and methods

Please describe the study setting and define the study population. 

Describe the study duration and the participants’ selection process (sampling technique)  

The study design looks more like a retrospective cohort. The authors should define the outcome measures in a separate section

It is good to use upper limit of reference range for the laboratory

Do you consider to include the clinical symptoms of hepatitis?

There are other adverse effects of isoniazid such as peripheral neuropathy, lupus and psychosis. Were these considered in this study? If they were not considered, the authors should provide an explanation to the audience. The authors should also mention how frequently these side effects were documented by the service providers.             

Results

The title for table 1 can be written as ‘Table 1: Demographic characteristics and outcome measures of latent TB treatments among Ethiopian Immigrants in Israel’

Any abbreviation in a table or figure must be explained beneath the table or figure. The authors should therefore add the meaning of EI. 

Table headings should be revised. For example, DOPT N (%) =231 (x%)

If the age range you used is less than 18 and 19-34, this means you may be missing someone who is 18 because 18 is not included in these age categories. Please address this. 

Please check the percentages in Table 1. Adding the total number of participants in the NOPT column gives a value of 100.1%.

Pruritus is non-specific feature and should be considered under this category

What is the denominator of the variables under the adverse effects. It is not clear what denominator is used in this calculation. 

The authors should add the confidence intervals to provide clarity as to whether these figures were truly statistically significant 

Discussion 

The discussion section needs a review. The authors should discuss the results using the STROBE approach; summarize the findings, interpret the result, discuss the generalizability and limitations of the results. 

The statement ‘Treatment completion did not differ significantly between three months; once weekly rifapentine isoniazid vs daily rifampin isoniazid regimens [9]. However, treatment completion was higher in the foreign-born group vs Norwegian-born. Foreign-born individuals in that study were more likely to be treated under direct observation and more commonly prescribed the rifapentine isoniazid regimen compared to Norwegian-born individuals, which may explain the difference between the two groups. The findings of Schein et al, are in contrast to earlier report regarding LTBI treatment completion in a cohort of 176 asylum seekers in Norway during 2005-2006, where only 1% completed treatment [10]’ is discussing another paper. The essence of discussing a paper is to compare your results with those of other studies. In many parts of the discussion section, you discussed previous research work, which is not appropriate. I suggest you address this right across the discussion section. 

There is no need to mention P value in the discussion section. Please remove them. They are already on the results.

Line 202: I don’t agree with the statement ‘The increased prevalence of mild GI and nonspecific complaints in the SAT group might be related to the inconvenience of the bus rides to the TB clinic’. First, you can't use the words mild GI and increased prevalence of nonspecific features. You have no baseline to compare with. Second, these thoughts are not scientific. If you attribute the high GI side effects to ‘bus ride’, how can you explain its high occurrence among people that are not doing the DOPT? 

Line 205 to 208: What the authors could have done was to compare the proportion of patients with elevated transaminases in the two groups. But it is unscientific to say that those in the SAT group had higher transaminase rates because more liver function tests were done in this group. 

Line 208: Please avoid words like ‘perplexing’ 

Lines 217 to 219: ‘There is a high prevalence of positive TST among EI’s; 37% 218 among those 19-34 years old and 41% among those older than 55 years [7]’. This data is not available in your result section and it is not appropriate to discuss the results of another study in this article. Please fix and consider deleting the whole paragraph

Line 227: Our study has several strengths… rather than using the word ‘advantage’

Lines 238 to 240: The statement ‘The 2.7% grade III isoniazid 238 related hepatitis in the SAT group underlines the need for a close follow-up examination to timely detect adverse drug reactions and prevent progression to serious liver injury’ refers to the implication of the findings and can be incorporated into the discussion or added in the conclusion. 

Conclusion

Not adequate. Please add the implications. 

Author Response

(The authors gave the same response as above.)

Round 2

Reviewer 1 Report

All my recommendations was accepted. 

Reviewer 2 Report

1. Please remove the P-values in the discussion section when you are proof-reading the paper. They are already in the result section

2. Table 1: Demographic characteristics and outcome measures of latent TB treatments among Ethiopian Immigrants in Israel SA. In this statement, remove the 's' from the word 'treatments' and add 'treatment'. 

3. I revised your conclusion to improve on the english. 'Nurse-administered isoniazid DOT for LTBI is an effective, safe, and cost-effective alternative to the SAT for high-risk, hard-to-reach new immigrants living in reception centers. Well-organized health services cannot only achieve high LTBI treatment completion rates, but also minimize the attrition throughout the LTBI cascade of care regardless of the treatment regimen'.